# The Role of the Mind-Body Connection in Children with Food Reactions and Identified Adversity: Implications for Integrating Stress Management and Resilience Strategies in Clinical Practice

**DOI:** 10.3390/children10030563

**Published:** 2023-03-16

**Authors:** Olivia Lee, Maria Mascarenhas, Robin Miccio, Terri Brown-Whitehorn, Amy Dean, Jessi Erlichman, Robin Ortiz

**Affiliations:** 1Perelman School of Medicine, University of Pennsylvania, Philadelphia, PA 19104, USA; olivia.lee1@pennmedicine.upenn.edu (O.L.);; 2Division of Gastroenterology, Hepatology, and Nutrition, Children’s Hospital of Philadelphia, Philadelphia, PA 19104, USA; 3Integrative Health Program, Children’s Hospital of Philadelphia, Philadelphia, PA 19104, USA; 4Division of Allergy and Immunology, Children’s Hospital of Philadelphia, Philadelphia, PA 19104, USA; 5Departments of Pediatrics and Population Health, New York University Grossman School of Medicine, New York, NY 10016, USA; 6Institute for Excellence in Health Equity, NYU Langone Health, New York, NY 10016, USA

**Keywords:** stress, adversity, resilience, integrative medicine, pediatric food intolerances, non-allergic food reactions, gastrointestinal

## Abstract

Child adversity is often associated with poor quality of life in pediatric gastrointestinal disorders, including non-allergic food reactions (food intolerances), which may be improved using mind-body interventions. We conducted an observational study to (1) describe child adversity (stressors) and resilience factors in children with food intolerances, and (2) explore the association between stressors and self-reported use of integrative modalities. A retrospective chart review of children ≥4-years-old presenting to a pediatric food intolerances clinic from 2017 to 2020 was performed (n = 130). Use of integrative medicine at intake, demographic, illness, and social history data were collected. Qualitative analysis identified exposure to stressors and resilience strategies. Correlation was assessed using a chi-square test. Management of the medical condition was the most common stressor, indicating impact on quality of life. Resilience strategies included themes of self-coping and social support. Individuals with one or more stressors were more likely to be using an integrative modality (most commonly, mind-body interventions) prior to their visit (*X*^2^ = 8.1, *p* = 0.004). Our hypothesis-generating study suggests that screening for child adversity and integrative medicine use may be used to better address quality of life and personalized approaches to treat pediatric food intolerances.

## 1. Introduction

Approximately 35% of children in the United States experience stress-related health problems [1]. Prolonged exposure to psychological stress and adversity in the absence of social and emotional buffers, referred to as “toxic stress”, is responsible for the biological effects of childhood adversity [2]; this may lead to mental, socio-emotional, behavioral, and physical conditions. Experiences associated with “toxic stress” may largely predict quality of life [3] and health outcomes specifically for those with gastrointestinal (GI) symptoms and disorders [4], evidencing the strong connection between the brain and gut (e.g., gut-brain axis).

Stress may manifest in GI somatic symptomology, eating and nutrition dysregulation, and underlying pathophysiological dysfunction that may lead to clinical GI disorders [5,6,7,8,9]. Though evidence suggests a close relationship between psychosocial stress and GI conditions, unexplained food reactions in general, and as potentially related to stress, remain understudied. Unexplained food reactions, or non-allergic food intolerances, which have been defined as non-immunological GI symptoms resulting from the pharmacological response to food components [10,11], are particularly common (15–25%) and underdiagnosed in children [12,13,14]. The diagnosis and treatment of unexplained food reactions is potentially challenged because such reactions may be in part related to experiences of stress; however, this remains understudied. However, if a relationship between stress and unexplained food reactions can be elucidated, the subsequent identification of stress mitigating, or protective and resiliency promoting, factors may facilitate the creation and study of enhanced treatment and intervention opportunities to support patients.

Resilience is the capacity to adapt to different circumstances, overcome adversity, and recover from challenges [15]. Resilience may manifest through an individual child, but is most often thought to be most strongly related to the presence of a supportive relationship in a child’s life (e.g., “relational health”), which effectively mitigates the effects of toxic stress [2]. The ability to rebound from stressors through self-resiliency qualities or mitigate the impacts of stress through the protective presence of relational health may result in maintained homeostasis and reduce physiological symptom burden in the short term [16]. Individuals with higher resilience tend to have lower risk of illness, accelerated healing, and a sense of well-being despite chronic illness [17]. Resilience aids in the functioning of the central nervous system [18], which also supports stress management. One aspect of the trait resilience may involve self-efficacy for coping strategies [19]. By learning and utilizing stress management techniques, patients are able to cope with both psychosocial stress and stress caused by management of and negative effects from their GI disorder. However, “resilience” in children may manifest through their utilization of aspects of their environment for emotional self-regulation and coping [20]; for example, engaging activities such as play and hobbies, or interacting with peers and supportive caregivers [2,21,22].

Integrative modalities such as mindfulness-based approaches can be used to build resilience [23] and have been shown to be effective in managing GI conditions [24,25] and reduce the adverse effects of childhood stress [26]. These integrative modalities include mind-body therapies (including supportive mental health interventions), acupuncture, diet, and some dietary supplements such as probiotics and herbs [27]. Other common integrative modalities used for resilience include yoga, massage, and cognitive-behavioral stress management techniques [28]. Any potential effectiveness of these integrative modalities in treating GI conditions may be due in part to the gut-brain axis. Psychological treatments such as cognitive behavioral therapy and hypnotherapy have also been found efficacious in treating GI disorders [29]. While coping and resilience strategies including the use of integrative modalities can mitigate GI symptoms, less is known about the prevalence of these factors in those who have experienced stressful events. Managing both psychosocial and nutritional components of care for children with unexplained food reactions is imperative to address the underpinning pathophysiological manifestations of the gut-brain axis [30], but this can only be done after descriptive analyses enhance the field’s understanding of the relationship between stress and integrative (such as mind-body) interventions.

Despite the established connections between stressors and GI conditions, little is known about the relationship between the use of integrative mind-body therapies, resilience strategies or coping experiences, and GI conditions. We aimed to describe both the contributions to the mind-body experience (e.g., stress and resilience) of children and their use of mind-body modalities, specifically in those presenting with unexplained food reactions to a Food Reactions Clinic (FRC) by: (1) quantifying stressors and resilience factors in this population, and (2) exploring the association between patient stressors and use of integrative modalities. This FRC is a unique clinic that specializes in unexplained food reactions. We hypothesize that (1) those presenting to the FRC will have a high prevalence for stressors, and (2) there is an association between stressors and the usage of integrative modalities. To date, the relationship between these specific stressors, resilience factors, and integrative modalities has not been studied in pediatric populations presenting with food reactions, but such work could inform how future studies aim to address and enhance quality of life for these patients.

## 2. Materials and Methods

A cross-sectional retrospective chart review of children who presented to a pediatric FRC at Children’s Hospital of Philadelphia (CHOP) from 2017 to 2020 was performed.

### 2.1. Clinical Setting

The clinic offers care to pediatric patients presenting with adverse reactions to food and GI symptoms despite testing negative for common food allergies who were referred to the clinic after an initial evaluation. The clinic consists of a multidisciplinary team specializing in allergy and immunology, gastroenterology, integrative medicine, and nutrition (inclusive of a registered dietician (RD)). Every patient sees a physician and has a full evaluation with an RD. Examples of recommendations from the RD may include specific diet plans, recipes, and plans for reintroduction of eliminated foods. In each physician’s one-hour visit, the physician conducts a detailed history of the present illness including a timeline from birth until the presenting concern, a physical exam, and then an assessment and treatment plan. Importantly, each clinic visit by physicians includes formally asking the use of integrative medicine modalities in the past 6 months, if known to the patient and/or accompanying caregiver. In conjunction with standard practice, patient-specific treatment plans may include integrative modalities such as nutritional guidance, stress-management, mindful movement (e.g., yoga), acupuncture, aromatherapy, or further evaluation if/as needed (e.g., laboratory testing, other sub-specialty referrals). We collected information on the number of patients who visited the clinic more than once (n = 51).

### 2.2. Patient Population

The clinic enrolls new pediatric patients (≤18-years-old) and continues to follow them indefinitely, with the oldest patient seen in April 2020 being 19-years-old. Inclusion criteria for our study were children ≥ 4-years-old at the most recent visit, based on the average developmental abilities of a 4-year old being closer to school age, namely their developing capacity to reason about choice and agency [31], which affect their ability to engage in strategies to regulate their responses to stressors [32]. Demographic, patient illness, social history and medical intervention data were collected using the Research Electronic Data Capture (REDCap) database system [33,34]. Of note, demographic data related to race were collected and quantified, but are limited to data in the electronic medical record and so may not be self-reported identifiers which are known have a high risk of inaccuracy [35]. Additional data were collected when available for a given patient chart, including antibiotic prescription patterns, sleep habits, hypermobility, and gestational age at birth, breastfeeding history, and screen time.

### 2.3. Stressors and Resilience

Patient electronic medical record data were specifically analyzed for content related to any history of exposure to stressors, and/or use of coping and resilience strategies, using a qualitative thematic content analysis approach as had been applied by team members prior [36] or those who visited more than once, our qualitative analysis included a review of notes from all visits to the FRC. Stressors were defined as any adverse childhood experience, significant adversities, and interpersonal problems in school and family environments [37]. This definition was based on experiences that are characterized as early life stress which are “conceptualized as childhood adversity, child maltreatment, or childhood trauma” [38]. Resilience strategies were defined to be the activities that patients engage for emotional regulation, personal enjoyment, or stress relief [15], with particular attention to mind-body practices that may be unique for children even if not commonly considered resilience strategies for adults (e.g., play, hobbies, quality time with caregivers) [39]. Content categories (codes) were defined for identified stressor and resilience strategy types. Code categorization was discussed by the first and senior author through reconciliation meetings to ensure a “continuous dialogue between researchers to maintain consistency of the coding” [40]. When codes were finalized, they were applied to the entirety of the sample of collected REDCap data by the first author. Resilience strategies were discussed among the study team and codes were abstracted up to themes.

### 2.4. Integrative Medicine Use

At each initial visit intake (first visit to the clinic), patients were uniformly asked by clinic staff (study team members) if they had used integrative medicine in the prior 6 months. Patients with prior use of integrative medicine were asked to specify which modalities were used. For analysis, integrative medicine modalities were then coded by the study team into subgroups defined by the NCCIH: psychological (e.g., mindfulness), physical (e.g., massage, acupuncture), and nutritional (e.g., dietary) interventions. Interventions were also categorized by mind-body practices, natural and products and nutritional modalities, and traditional healing approaches [27]. Correlation between stressors and integrative modalities was assessed using a chi-square test. Sensitivity analysis was done to correlate stressors with integrative modalities not including physical therapy, given physical therapy is commonly prescribed in a variety of allopathic clinical settings.

## 3. Results

### 3.1. Study Population Characteristics

There was a total of 134 patients seen at least once in the clinic between January 2017–April 2020. The majority (57%) of patients were seen only one time. The patient population during this time was 2- to 19-years-old, with 130 included in this study based on being 4-years and older. Subjects were mostly male (57.6%), white (75.4%), and of average age 10.36 (±4.50) years (Table 1). The majority (68%) of patients had more than one presenting symptom. The most common presenting symptoms were food intolerances (55%) and abdominal pain (48%) (Figure 1). Figure 1 displays the primary symptom presenting to the clinic where the x-axis displays the symptom, and the y-axis displays the number of patients with that symptom. The numbers reflected are inclusive of patients who presented with more than one of the symptoms displayed on the x-axis. The bottom value (dark blue) indicates male patients while the upper value (light blue) indicates female patients. Examples of “other” symptoms include migraines, hypotonia, and food protein induced enterocolitis syndrome.

### 3.2. Stressors

The majority (75%, n = 98) of participants had at least one stressor. A total of 10 codes were identified for types of stressors faced by children presenting to the FRC, as described in Table 2. The most common stressor was management of the medical condition and its impact on the patient’s daily life (51% of those with stressors, Table 2). The next most common was managing mental health. Examples of interpersonal stressors included experiencing bullying at school and conflict with family members. Environmental stressors included financial concerns, recent move/changing schools, and conflict between parental figures. Other common stressors were managing behavioral problems and learning difficulties at school.

### 3.3. Resilience

The majority (71%) of participants used at least one resilience strategy. A total of 13 codes were identified for types of resilience strategies used by children presenting to the FRC, as described in Figure 2. Figure 2 displays the resilience strategy along the y-axis and the number of patients using that strategy along the x-axis. The numbers reflected are inclusive of patients who presented with more than one resilience strategy. Majority (57%, n = 98) presented using more than one resilience strategy upon intake. Percentages represent the portion of those with resilience strategies (n = 98). The most common code included sports/physical activity which could have been done alone or with a group. However, the remaining frequently observed codes lead to two broad themes: self-coping and supportive relationships. Common resilience strategies (used by 10 or more patients) related to self-coping included listening to music, breathing and mindfulness activities, watching TV, reading, taking a bath or nap. The theme of supportive relationships was common (27%) and involved spending time with family and friends. Additionally, less common resilience strategies (used by less than 10 patients) involved nature, praying, artistic pursuits, video games, working, and seeing a therapist.

### 3.4. Integrative Modalities

Integrative modalities can be classified into categories of “mind and body practices”, “natural products” including nutritional interventions, and “traditional healing approaches” (Figure 3), informed by NCCIH [27]. Figure 3 displays the integrative modalities along the x-axis and the number of patients who use that modality on the y-axis. The numbers reflected are inclusive of patients who presented using more than one integrative modality. Mind body practices comprised the majority of integrative modalities, including physical therapy or occupational therapy, chiropractic, and prayer, psychological (including cognitive behavioral therapy), music, yoga, meditation, biofeedback, acupuncture, breathing, guided imagery, distraction, and massage. Natural products and nutritional interventions included nutrition and aromatherapy. Traditional healing approaches included holistic, naturopathic, and homeopathic interventions. Mind-body practices comprised the majority of integrative modalities used by patients. This category comprised a large variety of modality types, however, including musculoskeletal therapies (physical therapy or occupational therapy, chiropractic), prayer, yoga, meditation, and other (Appendix A). Natural products and nutritional modalities was the second most commonly used category of integrative medicine, but comprised only nutrition (e.g., dietary practices) and aromatherapy (Appendix A).

### 3.5. Asssociation between Stressors and Integrative Modalities

Of the patients with any stressor (n = 98), 59 (60%) were using integrative modalities. Those with no stressors were more likely to not be using integrative modalities at the time of intake to the FRC (those with one or more stressors who use integrative modalities vs. those with no stressors who use integrative modalities: 59 (60%), 10 (31%), X^2^ = 8.1, *p* = 0.004, Table 3). In sensitivity analysis, removing physical therapy, given that it is related to movement therapy, it is not necessarily considered an integrative modality, so the analysis did not change the observed results (Appendix A).

## 4. Discussion

Our data suggest children with non-allergic food reactions have a high occurrence of stressors. Furthermore, we observed a correlation between presence of stressors and usage of integrative modalities. The most common stressor was management of the medical condition, the most common resilience strategy was sports/physical activity, and the most common integrative modality category was mind-body. Our data suggest that children with stressors were more likely to report using integrative modalities in the past six months of the current visit. In patients with unexplained food reactions, future studies should aim to explore the extent to which mind-body integrative modalities are beneficial for patients’ stressors as well as for patients’ symptoms. Collectively, our results support the potential role of stress in non-allergic food reactions and related GI conditions, as well as the potential role for intervention through enhancing mind-body practices and resiliency (Figure 4). Figure 4 displays the potential role for mind-body resiliency and relational health in reducing toxic stress physiology that may be associated with GI conditions such as dysregulated eating behaviors, somatization, gut activity, permeability, and metabolic processes. Somatization is defined as unexplained somatic symptoms and may include GI or other system manifestations such as “stomachaches”, nausea or vomiting, dizziness, where these symptoms and constipation may also be associated with exposure to childhood adversity [41].

### 4.1. Literature, Gaps, and Opportunities

Psychosocial distress prevalent in children presenting with food reactions suggests perturbation of the brain-gut axis. This is corroborated by the literature which demonstrates an association with psychosocial stress at different levels of manifestation of symptoms and disorders in the GI system (Figure 4). For example, exposure to adversity, such as violence in a child’s community, and emotional responses to stress, such as anxiety, have commonly been associated with child reporting of somatic “stomachache” symptoms [5]. Exposures to environmental adversity or stress may also impact child behavior including eating patterns, such as increases in the consumption of unhealthy, high fat, foods, and snacking [6]. Furthermore, as exacerbated by the COVID-19 pandemic, parenting stress may translate into dysregulated feeding of children especially in using food as emotional reward [9]. This has the potential to establish and/or exacerbate children’s perception of the link between stress (child and caregiver) and food consumption, manifesting as problematic eating patterns, such as food fussiness and low food enjoyment [9]. Through brain-gut neurohormonal mechanisms, psychological stress has been shown to increase intestinal permeability [8]. For this reason psychosocial stress has been identified as a risk factor for development of functional GI disorders [7].

While our results support that stress inclusive of environmental stressors (e.g., caregiver/family stress or community adversity) may be prevalent enough to play a role in non-allergic food reactions for patients, importantly, patient experiences and presentations may be related to both the more removed external environmental stressors and proximal stress from the burden of the symptoms, quality of life limitations, and medical treatment. This is supported by our observation that the most common stress reported was related to the medical condition and its associated circumstances (e.g., doctor’s visits, testing). Similar findings are corroborated by the literature related to the condition ARFID or Avoidant/Restrictive Food Intake Disorder which, “may also result in significant impairment in personal, family, social, educational, occupational, or other important functioning (e.g., avoidance or stress related to participation in social eating experiences)” [42].

Stressors from interpersonal relationships, the household or family environment, community environment, and/or medical and healthcare experience can all contribute to the risk of chronic stress. Chronic stress may impact physiological balance in the stress pathway (hypothalamic-pituitary-adrenal axis) and related body systems (endocrine, immune, cardiometabolic), known as toxic stress [43,44]. It is known that both behaviors, such as those associated with non-allergic food reactions, for example, dysregulated eating, and toxic stress, potentially each through exposure to early childhood adversity, can increase risk for life-long morbidity and mortality [45,46]. Importantly, the perturbations manifested in the stress axis due to chronic stress have more acute, or immediate, impacts on signaling at various levels of the GI tract [47].

Resilience and perceived self-efficacy, the perception of being able to problem-solve, are important for children’s ability to navigate challenges in everyday life [19], and may be relevant for the stressors seen in our patient population. These adaptive coping mechanisms of resilience allow the individual to avoid the negative consequences of stress that would otherwise negatively impact their physiological or physical well-being [18]. Factors that can promote resilience against the manifestations of disease through mitigating toxic stress are hypothesized, based on the literature, include healthy lifestyle behaviors (e.g., physical activity and eating Mediterranean dietary patterns of eating), exposure to nature, mindfulness, access to mental and physical healthcare, and, most importantly, safe, stable, and nurturing relationships [2,48]. Clinically, implementation of such practices may involve harnessing integrative modalities that promote many of these pillars simultaneously (e.g., mindfulness family practices, or nature immersion). What may be more known intuitively to patients as resilience strategies may be as simple as play and hobbies, which can be supported through engaging formal integrative medicine practices, or less necessarily clinical approaches to supporting healthy dyadic (caregiver-child) and family relationships, both toward the reduction of stress. It is possible that future studies of interventions to support those with high stress and comorbid GI conditions (i.e., patients in this FRC) needs to consider harnessing both integrative modalities and resilience to mitigate stressors.

Studies have shown that self-coping activities (i.e., listening to music, spending time in nature, breathing, and mindfulness exercises), as well as socially supportive activities (i.e., sports, quality time with friends and family, prosocial hobbies), are significant predictors for resilience, improve children’s perceived well-being, and reduce negative impacts of stress [21,49]. For example, children who participated in an after-school yoga program that met for just one hour per week for twelve weeks reported using fewer negative behaviors in response to stress [49]. Additionally, physical activity (performed alone or in a group) has been shown to not only reduce stress, but also to improve GI symptoms in patients with irritable bowel syndrome [50,51]. Resilience factors unique to children, including play and hobbies, mitigate the mind-body challenges of stress [39]. Furthermore, mindfulness itself has been identified to have a role in reducing the impact of childhood adversity and stress on numerous conditions in childhood and into adulthood [26].

The literature on integrative modalities and stress has largely focused on diet and mindfulness interventions. Both of these interventions have proven to mitigate stress or stress-physiology (“toxic stress”) [52], and potentially improve symptoms of GI conditions. For example, studies have been done on the impact of integrative therapies on specific conditions such as diet therapies on Crohn’s disease [25], autism [53], depression [54], and GI symptoms. For GI symptoms, there is evidence that acupuncture, diaphragmatic breathing exercises, and a specific diet (reducing certain foods) may reduce gastroesophageal reflux disease (GERD) symptoms, while nutritional interventions (ginger) and aromatherapy have been shown to reduce nausea and vomiting in adult populations [24]. Mindfulness has been shown to reduce stress, anxiety, and improve markers of health such as chronic pain [52]. There is need for investigation of the effects of the combined effects of resilience factors inherent to children (i.e., play, hobbies) and formal integrative medicine (i.e., mindfulness-based stress reduction) techniques.

This study has some limitations. First, this is a retrospective chart review study and so there may be a risk of underestimation of patient stressors, given the lack of standard screening in clinic visits. However, this is likely balanced out by the potential risk of overestimation of stressors due to researchers seeking out this information from the chart. Importantly, while our definition of stressors was intentionally broad for the hypothesis-generating nature of this study, the use of questionnaires with patients in future studies will enable the differentiation of types of adversity (e.g., adverse or expanded childhood experiences [37], other trauma, or other experiences), noting the inclusion of screening for positive childhood experience can expand capture of potential resilience factors [55,56]. Second, there may be bias specific to certain provider practices in the FRC. While it is standard for physicians in the practice to take timelines of illness for each patient, there was not a standardized format that the history was recorded. Additionally, the patient population was mostly racially and ethnically homogenous, so future work should seek to explore diverse populations. Due to the retrospective nature of the data, we were limited to finding correlations and prevented from finding causations. However, the study remains strong in identifying a correlation between stressors and patient-reported integrative medicine use, demonstrating a need for further research to investigate if integrative modalities improve symptoms of stress and unexplained food reactions.

### 4.2. Future Directions

Such studies may be designed for prospective data collection to measure potential enhancement of children’s existing resilience strategies by the use of integrative medicine. With this approach, future work can be done to determine whether stressors and/or resilience factors predict outcomes or responses to prescribed integrative modalities, inclusive of techniques that directly address the mind-body connection and allow physicians to identify which patients might benefit from certain interventions. Since the mind-body connection is crucial in children with non-allergic food intolerances, improved understanding of mind-body integrative interventions is needed to best serve this population. It is important to note that our study identified that the category of mind-body modalities was the most common, however it was inclusive of numerous (14) types of mind-body practices. Therefore, future research should aim to explore which of these modality types may be most effective for children with high stress and/or non-allergic food reactions. Further, the second most common category was nutritional modalities (e.g., nutrition and aromatherapy). Therefore, other work should consider the potential for enhancing efficacy of interventions for non-allergic food reactions by combining mind-body with nutritional modalities.

Given the high rate of stressors identified in patients with non-allergic food intolerances, a prospective study may aim to incorporate screening for child adversity to better understand its relationship to non-allergic food reaction symptoms, treatment, and clinical response. Such studies may consider using the Pediatrics adverse childhood experiences and related life events screener (PEARLS) [57], or Childhood Trauma Questionnaire (CTQ) [58], for example. This screening should also include quality of life and disease burden measures to help elucidate the role of the proximal stress created by the lived experience of the condition (e.g., medical care, social impacts of the condition), versus the role of external and environmental stressors (e.g., adverse childhood experiences in the home or caregiver environment) to best target how to improve patient quality of life. Furthermore, given the known association between caregiver stress and feeding practices of children [9], it may be important to directly measure caregiver-reported stress.

Despite the need for future research to more accurately measure stress and adversity in pediatric GI studies, population-level research does not translate into individual-level clinical practice. Future work must take care to elucidate if there is any clinically meaningful utility in screening for psychosocial stress or childhood adversity before considering this implementation. Screening for stressors without promoting protective factors or offering supportive environments to foster self, dyadic, and family resiliency, may risk triggering or re-traumatizing patients who have experienced adversity [59,60]. However, even more upstream of screening practices, there is an opportunity to establish trauma-informed environments and care practices in all clinical settings. Such a model may be adapted from that of the Substance Abuse and Mental Health Services Administration [61], to involve the realization about how stress and trauma may impact patients and caregivers (including healthcare providers), recognition of the symptoms of stress and trauma (and specific GI and quality of life manifestations), responding to stress and trauma (which may, in part, leverage integrative medicine models and modalities), and resisting re-traumatization (by creating safe and trusting clinical environments and communities) (Figure 5). Figure 5 illustrates the “4 R’s” of Trauma Informed Care (adapted from the Substance Abuse and Mental Health Services Administration) to address Adverse Childhood Events (ACES) and treat toxic stress in GI conditions: Realize, Recognize, Respond, and Resist.

In conclusion, stressors of various origins can manifest as GI dysfunction, which may provide context as to why we observed a strong prevalence of stressors in a cohort of pediatric patients with non-allergic food reactions. Given these findings, it may be important for clinicians who see pediatric patients with GI conditions to practice trauma informed care by centering the experiences, and resiliency strategies and factors, of children and their families. Additionally, it is imperative that future work aims to better understand the relationship between toxic stress, resilience and integrative medicine use and response in children with non-allergic food reactions.

## Figures and Tables

**Figure 1 children-10-00563-f001:**
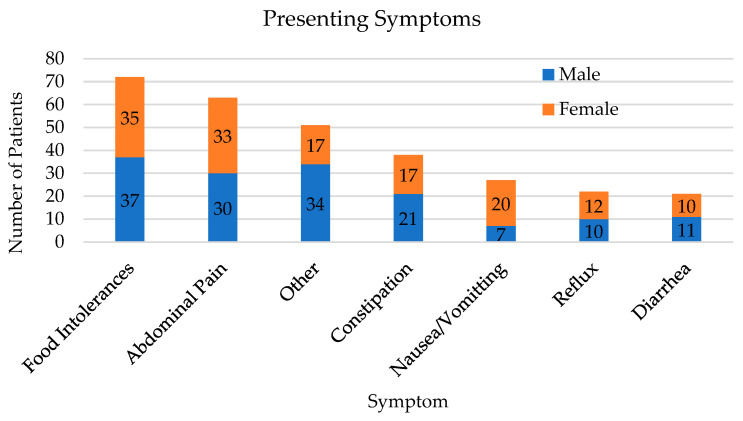
Presenting Symptoms.

**Figure 2 children-10-00563-f002:**
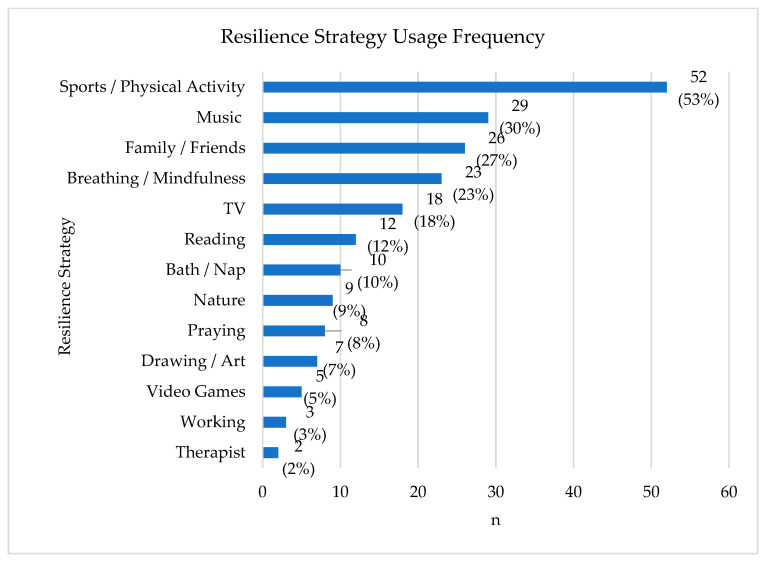
Resilience Strategies.

**Figure 3 children-10-00563-f003:**
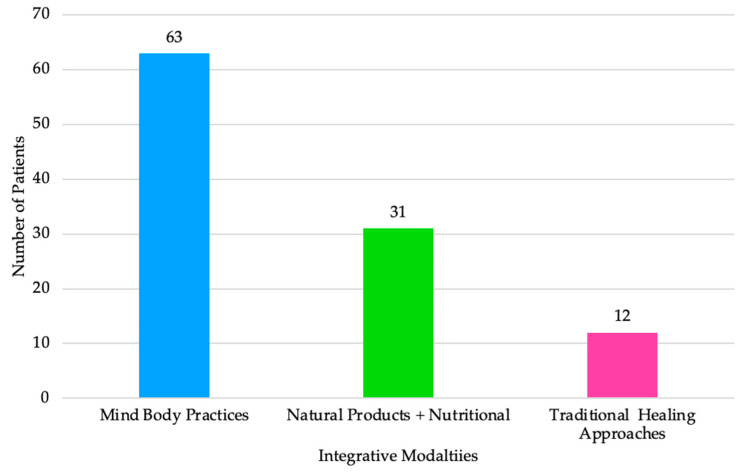
Number of Patients Using Integrative Modalities.

**Figure 4 children-10-00563-f004:**
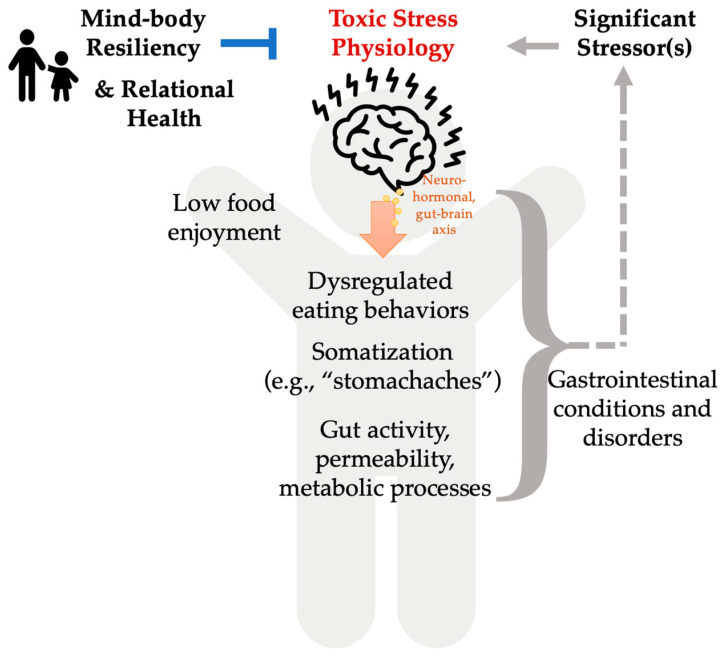
Hypothesized Toxic Stress Manifestations in GI Disease, Resiliency, and Relational Health.

**Figure 5 children-10-00563-f005:**
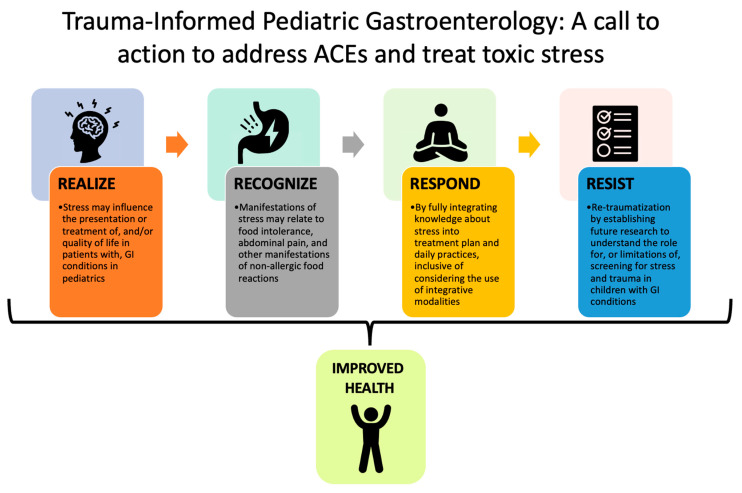
Trauma-Informed Pediatric Gastroenterology: A call to action to address ACEs and treat toxic stress.

**Table 1 children-10-00563-t001:** Demographics of patient population presenting to FRC (N = 130).

Variable	Variable Category	Participants by Category, n (%, N = 130)
Total Sample	N = 130	
Gender	Male, n (%) Female, n (%)	74 (57%) 56 (43%)
Average Age *		10.36 years
Race	White, n (%) Asian, n (%) Black or African American, n (%) Unknown or Not Reported, n (%)	98 (75%) 10 (8%) 5 (4%) 17 (13%)
Ethnicity	Not Hispanic or Latino, n (%) Hispanic or Latino, n (%) Unknown or Not Reported, n (%)	125 (96%) 3 (2%) 2 (2%)
Stress, Resilience, and Integrative Modality Use	Participants with Stressors, n (%) 1 stressor, n 2 stressors, n 3 stressors, n Participants with Resilience Strategies, n (%) Participants with Integrative Modalities Upon Intake, n (%) Participants with Integrative Modalities Prescribed, n (%)	98 (75%) 58 34 6 98 (75%) 69 (52%) 110 (85%)

* Given some patients were seen in the clinic for follow-up visits, average age was calculated based off their age at the first presentation to the clinic.

**Table 2 children-10-00563-t002:** Identified Stressors.

Code	n (%, N = 98)	Chart Example
Medical	50 (51%)	“Doctors’ visits and tests” “GI symptoms” “Medical problems (limited diet and being sick often)”
Mental Health	20 (20%)	“After school tends to be tough emotionally (weekends differ)” “Reports difficulty relaxing—stressors are his anxiety and OCD”
School—Other	19 (19%)	“Change of school” “School does not see as an important health issue that needs accommodation”
Family—Other	14 (14%)	“Hates to be separated from mom” “Conflict with sister and aggression; noncompliance”
School—Learning	11 (11%)	“Poor performance in school [is causing stress]” “Learning disability, anxiety, lack of focus, hyperactivity, self-stimulatory activities [while in school]”
Family—Parents	11 (11%)	“Parenting differences—dad is in a lot of denial about the symptoms they’ve had to deal with for the past 6–7 years, mom is actively trying to recover her own health, they are in counseling to work through issues and asked for guidance to get their daughter to the next stages” “Volatile relationship with father”
Behavioral	9 (9%)	Mom reports “impulsive, hyperactive, eating issues, wet bed, emotion control, attention issues”
School—Bullying	5 (5%)	“Bullying at school a little” “Change of school, bullying”
Financial	2 (2%)	“College and lack of a job”
Recent Move	2 (2%)	“Stressors: change of school, recent family move” “Recent family move”

**Table 3 children-10-00563-t003:** Correlation between Presence of Stressors and Usage of Integrative Modalities.

	Stressors—Y (n = 98)	Stressors—N (n = 32)
Integrative Methods—Y	59 (60%)	10 (31%)
Integrative Methods—N	39 (40%)	22 (69%)

## Data Availability

Data can be requested from the corresponding author.

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
