# Peer review of "The Role of the Mind-Body Connection in Children with Food Reactions and Identified Adversity: Implications for Integrating Stress Management and Resilience Strategies in Clinical Practice"

_children, 2023, doi:10.3390/children10030563_

Round 1

Reviewer 1 Report

1) Minor language issues :

- line 60-61 aids to (used twice in sentence)

- line 73-76 in part (used twice)

- line 93 to colloquial language

- line 103 (better to use passive voice)

- line 122 acronym REDCap should be explained

- 149-152 the method is uncleare should be explained in further detail

Author Response

1) Minor language issues:

REPLY: We thank you for your review and were happy to address the suggested comments as below.

Line 60-61 aids to (used twice in sentence)

REPLY: Before revision, line 60-61 read, “Resilience aids in the functioning of the central nervous system[16], which also aids stress management.” After revision, lines 67-68 read, “Resilience aids in the functioning of the central nervous system[18], which also supports stress management.”

Line 73-76 in part (used twice)

REPLY: The sentence, “Any potential effectiveness of these integrative modalities in treating GI conditions may be in part due in part to the gut-brain axis” has been revised to now read, “Any potential effectiveness of these integrative modalities in treating GI conditions may be due in part to the gut-brain axis.”

Line 93 to colloquial language

REPLY: Before revision, line 99 read, “To date, the relationship between these specific stressors, resilience factors and integrative modalities has not been studied in pediatric populations presenting with food reactions, but such work could inform how future studies aimed at addressing and enhancing quality of life for these patients.” It has been updated for clarity and now reads, “To date, the relationship between these specific stressors, resilience factors and integrative modalities has not been studied in pediatric populations presenting with food reactions, but such work could inform how future studies aim to address and enhance quality of life for these patients.”

Line 103 (better to use passive voice)

REPLY: Thank you for this comment. In reviewing line 103 and the content of the paragraph surrounding it, we feel the language and style flow well for the reader. However, if the reviewer or editorial team would like to suggest particular wording or language to modify this line or paragraph, we would be happy to make such changes.

For your reference, line 133 (previously line 103) and its surrounding content are included below:

The clinic consists of a multidisciplinary team specializing in allergy and immunology, gastroenterology, integrative medicine, and nutrition (inclusive of a registered dietician (RD)). Every patient sees a physician and has a full evaluation with an RD. Examples of recommendations from the RD may include specific diet plans, recipes, and plans for reintroduction of eliminated foods.

Line 122 acronym REDCap should be explained

REPLY: We appreciate the suggestion for clarification. Previously line 122 read, “Demographic, patient illness, social history and medical intervention data were collected using the REDCap database system [30,31].” Line 154 now includes the acronym REDCap: “Demographic, patient illness, social history and medical intervention data were collected using the Research Electronic Data Capture (REDCap) database system [33,34].”

149-152 the method is uncleare should be explained in further detail

REPLY: Lines 149-152 previously read, “For analysis, integrative medicine modalities were then coded by the study team as those defined by the NCCIH including psychological (e.g., mindfulness), physical (e.g., massage, acupuncture), and nutritional (e.g., dietary) interventions and then codes were categorized by mind-body practices, natural and products and nutritional modalities, and traditional healing approaches [25].” After revision, lines 189-193 is as follows, “For analysis, integrative medicine modalities were then coded by the study team into subgroups defined by the NCCIH: psychological (e.g., mindfulness), physical (e.g., massage, acupuncture), and nutritional (e.g., dietary) interventions. Interventions were also categorized by mind-body practices, natural and products and nutritional modalities, and traditional healing approaches [27].”

Reviewer 2 Report

Non-allergic food intolerance is a common pediatric disease, but remains poorly understood and understudied. The authors studied the pediatric patients with this disease based on the data collected from 2017-2020 in the FRC of CHOP. The studies are well designed and presented. The conclusions regarding several interventions are convincing and important. This type of investigation is lacking in both clinics and research. I highly recommend acceptance of this article for publication.

Author Response

Non-allergic food intolerance is a common pediatric disease, but remains poorly understood and understudied. The authors studied the pediatric patients with this disease based on the data collected from 2017-2020 in the FRC of CHOP. The studies are well designed and presented. The conclusions regarding several interventions are convincing and important. This type of investigation is lacking in both clinics and research. I highly recommend acceptance of this article for publication.

REPLY: We appreciate your recognition of the importance of this work, and we thank you for your review. We also hope to see this article published with the support of you, the other reviewers, and the editorial team. Thank you!

Reviewer 3 Report

In this study Authors have done an observational study to describe child adversity (stressors) and resilience factors in children with food intolerances. Authors have also explored the association between stressors and self-reported use of integrative modalities. Overall the study is very interesting and given very limited studies describing stressors, resilience factor sand integrative modalities in pediatric patients the study is very informative about ways to enhance quality of life for these patients. 

I have brief comment to clarify before recommending this for editorial decision:

In the study it has been mentioned that the Age of participants was between 4-19 years old. Results mentioned are based on the Average age of 10.36. It will be interesting to highlight how stressors and integrative modalities differ if we take samples from two extreme age groups (very young compared to older).

Author Response

In this study Authors have done an observational study to describe child adversity (stressors) and resilience factors in children with food intolerances. Authors have also explored the association between stressors and self-reported use of integrative modalities. Overall the study is very interesting and given very limited studies describing stressors, resilience factor sand integrative modalities in pediatric patients the study is very informative about ways to enhance quality of life for these patients. 

I have brief comment to clarify before recommending this for editorial decision:

In the study it has been mentioned that the Age of participants was between 4-19 years old. Results mentioned are based on the Average age of 10.36. It will be interesting to highlight how stressors and integrative modalities differ if we take samples from two extreme age groups (very young compared to older).

REPLY: Thank you so much for recognizing the importance of age in the context of our study. We agree age may play an important role in considering developmental abilities of children, which is why we selected for a sample of children who were age 4 or older by the most recent visit. To address your important comment, we conducted an addition analysis to, specifically, see if there were differences in the most common stressor by age “extremes”.  We first defined “extremes” by quartiles (lowest and highest; <7 years and >14 years, respectively), and we then discovered that the most common stressor (management of the medical condition and its impact on daily life) remained the same for both age extremes. The most common integrative modality category (mind-body) remained the same for the youngest and oldest (age extremes) group(s). While we agree this would be an important area for future research, potentially through the collection of prospective data powered for comparisons by age, we felt adding this analysis by age in the current manuscript may not add to our study or findings.

Round 2

Reviewer 3 Report

Authors have addressed my comment. I recommend this for publication in present form.